# Investigation of the Properties of 316L Stainless Steel after AM and Heat Treatment

**DOI:** 10.3390/ma16113935

**Published:** 2023-05-24

**Authors:** Patrik Petroušek, Tibor Kvačkaj, Jana Bidulská, Róbert Bidulský, Marco Actis Grande, Diego Manfredi, Klaus-Peter Weiss, Róbert Kočiško, Miloslav Lupták, Imrich Pokorný

**Affiliations:** 1Department of Plastic Deformation and Simulation Processes, Institute of Materials and Quality Engineering, Faculty of Materials, Metallurgy and Recycling, Technical University of Kosice, Park Komenského 11, 04001 Kosice, Slovakia; robert.kocisko@tuke.sk (R.K.); miloslav.luptak@tuke.sk (M.L.); imrich.pokorny@tuke.sk (I.P.); 2Bodva Industry and Innovation Cluster, Budulov 174, 04501 Moldava nad Bodvou, Slovakia; tibor.kvackaj@tuke.sk (T.K.); robert.bidulsky@gmail.com (R.B.); 3Department of Applied Science and Technology (DISAT), Politecnico di Torino, Viale T. Michel 5, 15121 Alessandria, Italy; marco.actis@polito.it; 4Department of Applied Science and Technology (DISAT), Polythecnic of Turin, Corso Duca degli Abruzzi 24, 10129 Torino, Italy; diego.manfredi@polito.it; 5Institute for Technical Physics, Karlsruhe Institute of Technology (KIT), 76344 Eggenstein-Leopoldshafen, Germany; klaus.weiss@kit.edu

**Keywords:** additive manufacturing (AM), powder metallurgy, laser powder bed fusion (L-PBF), 316L stainless steel, cryogenic treatment

## Abstract

Additive manufacturing, including laser powder bed fusion, offers possibilities for the production of materials with properties comparable to conventional technologies. The main aim of this paper is to describe the specific microstructure of 316L stainless steel prepared using additive manufacturing. The as-built state and the material after heat treatment (solution annealing at 1050 °C and 60 min soaking time, followed by artificial aging at 700 °C and 3000 min soaking time) were analyzed. A static tensile test at ambient temperature, 77 K, and 8 K was performed to evaluate the mechanical properties. The characteristics of the specific microstructure were examined using optical microscopy, scanning electron microscopy, and transmission electron microscopy. The stainless steel 316L prepared using laser powder bed fusion consisted of a hierarchical austenitic microstructure, with a grain size of 25 µm as-built up to 35 µm after heat treatment. The grains predominantly contained fine 300–700 nm subgrains with a cellular structure. It was concluded that after the selected heat treatment there was a significant reduction in dislocations. An increase in precipitates was observed after heat treatment, from the original amount of approximately 20 nm to 150 nm.

## 1. Introduction

Additive manufacturing (AM), also known as 3D printing, is a process that involves building a three-dimensional object by adding material layer by layer [1,2,3,4]. In the case of metals, AM is typically performed using a laser powder bed fusion process (L-PBF), where a laser or electron beam is used to selectively melt the metal powder to create the desired shape. 316L stainless steel is a commonly used material in AM due to its excellent corrosion resistance, high ductility, and good strength at elevated temperatures [5,6]. However, the microstructure of 316L stainless steel produced using AM can differ from conventionally manufactured 316L stainless steel due to the rapid solidification rates and thermal cycling that occurs during the printing process [7,8]. The microstructure of AM-produced 316L stainless steel typically consists of columnar grains that grow vertically from the substrate and then laterally as the melt pool moves. These grains are often larger than those found in conventionally manufactured 316L stainless steel. The melt pool boundaries can also contain various defects such as porosity, lack of fusion, and cracks [9,10,11]. Additionally, the solidification conditions in AM can lead to the formation of different phases in the microstructure of 316L stainless steel. For example, the formation of the delta ferrite phase can be promoted due to the high cooling rates, which can affect the mechanical properties of the material [12]. The presence of these phases can be controlled by adjusting the process parameters, such as the laser power and scanning speed [13,14,15,16]. Understanding and controlling the microstructure of AM-produced 316L stainless steel is essential for optimizing its mechanical properties and ensuring its suitability for various applications [9,17].

In the International Thermonuclear Experimental Reactor (ITER), 316L stainless steel is used in various components, such as the vacuum vessels, divertor, and blanket modules [18,19,20,21,22]. AM techniques, such as laser powder bed fusion (L-PBF) and electron beam melting (EBM), are used to prepare the 316L-stainless-steel components for ITER [23,24].

The mechanical properties of 316L stainless steel prepared using L-PBF can vary depending on the specific processing parameters used. Generally, the material exhibits improved strength, ductility, and fracture toughness compared to conventionally manufactured 316L stainless steel [25,26,27,28,29,30,31,32]. Studies have shown that the tensile strength of 316L stainless steel prepared using L-PBF can range from 750 MPa to 1000 MPa, while the elongation at break can be as high as 50% [33,34,35]. Similarly, the yield strength (YS) of 316L stainless steel prepared using EBM can range from 590 MPa to 710 MPa, while the ultimate tensile strength (UTS) can be as high as 800 MPa [36,37]. In addition to the tensile properties, the fracture toughness of 316L stainless steel prepared using additive manufacturing has also been studied. The results indicate that the material exhibits improved fracture toughness compared to conventionally manufactured 316L stainless steel [38].

The formation of subgrains and dislocations in 316L stainless steel prepared using L-PBF can have significant effects on the mechanical properties of the material. For example, the presence of subgrains can lead to improved ductility and toughness, while the presence of dislocations can result in increased strength but reduced ductility. The size and shape of the cells in the cellular dislocation substructure can also affect the material’s properties. For example, smaller cells may lead to higher strength but lower ductility, while larger cells may lead to higher ductility but lower strength [39,40].

The formation of the cellular dislocation substructure in 316L stainless steel prepared using L-PBF is influenced by the processing parameters, such as the laser power, scanning speed, and layer thickness. Studies [41,42,43] have shown that increasing the laser power or decreasing the scanning speed can lead to the formation of finer cellular dislocation substructures, while increasing the layer thickness can lead to the formation of coarser cellular dislocation substructures.

Overall, dislocations can affect the mechanical properties of the material and can be mitigated through post-processing techniques, such as hot isostatic pressing (HIP) and heat treatment. Understanding the formation and characteristics of this substructure is crucial for optimizing the processing parameters and designing components with the desired properties [25,44,45,46].

L-PBF generally creates a high residual stress state in the produced components, resulting in distortion and warping, and leading to a variation in the final mechanical properties. Stress relieving, usually performed right after the L-PBF processing, helps in reducing residual stresses, increasing ductility, and decreasing tensile strength. These mechanical properties are very closely connected with the structural properties and therefore it is necessary to investigate these relationships for application outputs [34,41,42,43].

This study presents the development of the structure after L-PBF and subsequent heat treatment. The development and progress of the reduction of the number of dislocations and the development of precipitates after the selected heat treatment are analyzed. To evaluate the mechanical properties, a static tensile test was carried out at temperatures of 8K, 77K, and ambient temperature. After L-PBF, structural analyses were carried out on the material as-built and after selected heat treatment using optical microscopy, scanning electron microscopy, and transmission electron microscopy.

## 2. Materials and Methods

### 2.1. Materials

The material used for the experiment was the commercially produced as powder material, EOS Stainless Steel 316L (Fe—62.4 wt.%; C—0.08 wt.%; Cr—17.85 wt.%; Ni—14.59 wt.%; Mo—2.86 wt.%; Mn—1.55 wt.%; Si—0.47 wt.%; other elements—0.28 wt.%).

The particles of the experimental powder were analyzed using a FESEM Zeiss Supra 40 (Zeiss, Oberkochen, Germany) microscope. Figure 1 shows that most of the particles had a globular shape and a smooth surface. Moreover, nearly no agglomerates were present. The size distribution was monomodal and quite small, with diameters from around 13 µm to 57 µm, and 90% had a smaller diameter than 42 µm. These properties led to a high packing efficiency, good flowability, and the ability to be easily and homogeneously distributed into a flat and thin layer, which makes it a perfect fit for the L-PBF process. Since the powder material has a monomodal distribution of particle size, it can be assumed that the used powder contained neither significant contaminants nor foreign particles. A detailed analysis of the investigated material with a powder size distribution analysis was published in our previous work [47].

### 2.2. Production of Samples Using L-PBF Technology

The samples for the entire range of experiments were made using L-PBF technology on an EOS M280 Dual Mode (EOS GmbH, Krailling/Munich, Germany). The machine EOS M280 is equipped with a 200 W Yb fiber laser. The material was processed in an Ar atmosphere with a layer thickness of 20 µm and a meander scanning strategy. In this strategy, the scanning direction is rotated by 67° after each application. Table 1 shows the basic parameters of the manufacturing process, where P is the laser power, $S_{S}$ is the scanning speed, $h_{d}$ is the distance between the individual laser paths, and $S_{1}$ is the length of the welding strip.

### 2.3. Heat Treatment and Microstructure

The samples produced for the static tensile test and microstructure analysis were divided into two sets. The first set of samples was labeled “HT0”, which means that the samples were without heat treatment directly after processing (as-built). The second set of samples, marked as “HT2”, was selected for solution annealing at 1050 °C and 60 min soaking time, followed by artificial aging at 700 °C and 3000 min soaking time. Heat treatment was carried out in a vacuum furnace TAV MiniJet HP 235 (TAV, Caravaggio, Italy) with a vacuum level approaching $9\times 10^{-3}$ mbar. Due to the low temperature employed, there was no need for Ar backfilling to prevent the degassing of elements with low vapor tension. A microstructural analysis was performed on the samples using a standard metallographic procedure. After grinding and polishing, the microstructure was etched with aqua regia to observe the microstructure using optical microscopy (OM), scanning electron microscopy (SEM), and transmission electron microscopy (TEM). The samples for all the types of structural analyses were taken perpendicular to the longitudinal axis of the materials under investigation.

SEM and electron backscatter diffraction (EBSD) analyses were performed on a Tescan Lyra 3.

The TEM microscopes JEOL JEM 2200FS (JEOL, Tokyo, Japan), with a power of 200 kV, and FEI TECNAI, with a power of 200 kV, and equipped with energy-dispersive X-ray analysis (EDS), were used for TEM analysis. The analyzed samples for TEM were taken in the transverse direction to the built direction. The samples were ground to a final thickness of 50–70 µm. Electrolytic polishing was carried out on a Fishione twin jet with an electrolyte 5% solution of 70% HClO_2_ diluted in methanol.

### 2.4. Static Tensile Test

A static tensile test was carried out at three different temperature conditions. The static tensile tests at 77 K and 8 K were performed on an MTS 100 Landmark (MTS, Eden Prairie, Minnesota, MN, USA) equipped with a cryostat and an extensometer. The static tensile tests were performed in accordance with the standards for tensile [48,49]. A touch extensometer was used to measure deformations. The sample was cooled to 8 K in two stages. In the first stage, the sample was cooled to 77 K in liquid nitrogen. In the second stage, the sample was cooled in a cryostat with a liquid helium sheath. The static tensile test at ambient temperature was performed on a TINIUS OLSEN H300KU (Tinius Olsen, Salfords, England) machine. For all states, the strain rate during tests was $0.0023\;s^{-1}$.

## 3. Results

### 3.1. Static Tensile Test

Tensile tests were carried out at an ambient temperature of 298 K and at cryogenic temperatures of 77 K and 8 K for the states HT0 and HT2, and are shown in Figure 2. The cryogenic test at 77 K was performed in liquid nitrogen. For the 8 K tests, an evaporating gas cryostat was used and cooled with liquid helium. The L-PBF specimens at room temperature showed a large decrease in yield strength between HT0 and HT2. However, the strain hardening of HT2 resulted in an equal ultimate tensile strength as that of HT0. Investigations are still ongoing to determine the reason for the poor strain hardening of HT0. Additionally, HT2 exhibits an even greater uniform elongation than conventionally manufactured materials. The mechanical behavior of HT0 changes significantly at 77 K. At 77 K, HT2, as well as traditionally manufactured materials, exhibits strain hardening and uniform elongation. In summary, the mechanical properties of HT2 are in the same range as traditional manufactured materials, making the L-PBF process, with an adequate post-heat treatment, a viable option for applications under static loads. Disregarding the ductility at room temperature and at 8 K, even the yield and ultimate tensile strength of the as-built (HT0) specimens are in a similar range to traditional manufactured materials. A detailed analysis of the mechanical properties of the material was analyzed in our previous research [33,47].

### 3.2. Microstructure Analysis of the as-Built Samples

A typical material structure after L-PBF is shown in Figure 3. Visible weld strips copy the movement strategy of the laser. Cellular (Figure 3b—area A) and columnar (Figure 3b—area B) dislocation subgrains are visible in the OM images, which are the result of the laser action and the combination of the rapid heating and rapid cooling processes. This thermal event results in the accumulation of high values of internal thermal stresses in the materials directly after L-PBF, which results in a high proportion of dislocations at the subgrain boundaries. These substructures are separated by high-angle grain boundaries. No internal defects, discontinuities, or large irregular pores are visible on the OM images. Porosity below 0.1% has almost no effect on the resulting mechanical properties [8,11,47,50,51,52]. The pores that were detected and analyzed were characterized by an optimal spherical shape without significant sharp edges. The average grain size after L-PBF was D=25 µm.

The structural observations made using scanning transmission electron microscopy (STEM) showed that the subgrains produced by the L-PBF technology could have a cellular shape resembling “honeycombs” or a columnar arrangement elongated in one direction. The cellular arrangements of the subgrains are shown in Figure 4a. The columnar arrangement can be seen in Figure 4b. The subgrains are visible in Figure 4a and were formed by significant clusters of dislocations, which are typical for as-built materials after L-PBF. The subgrains may have a larger single dimension in one axis. Each subgrain may have a different shape depending on the number of neighboring subgrains. If one subgrain is surrounded by six others, it takes the shape of a hexagon. Columnar dislocation subgrains have a significantly larger length-to-width ratio. Columnar dislocation subgrains are a minor component of the subgrains compared to cellular dislocation subgrains.

Based on the measurements of the cellular dislocation subgrains, their size was in the range $\langle 300;\;700\rangle \;\left[\mathrm{nm}\right]$ (Figure 5a). For the columnar dislocation subgrains, the width is particularly important. The width of these dislocation subgrains was in the interval $\langle 300;\;900\rangle \;\left[\mathrm{nm}\right]$ (Figure 5b).

The TEM and STEM images indicate that the individual grains are composed of subgrains separated by a dislocation substructure. A large number of dislocations and particles can be seen inside the individual columnar dislocation subgrains. Individual subgrains that form one grain have the same orientation. The matrix was determined as γ-Fe using electron diffraction (Figure 6). The material contains a large number of small particles with a size of up to ~20 nm and a smaller number of large particles with a size of ~100 nm, which are marked by red arrows (Figure 6b). Most of the particles are randomly distributed inside the individual subgrains.

The element map in Figure 7 reveals the segregation of Cr at the cell wall, which seems to have a slight enrichment of Fe and Ni elements, indicating that the solid solution elements were redistributed during the formation of the cellular structure.

An EBSD analysis was performed on the samples which were as-built. There is an inverse pole figure in the direction of the *Z*-axis in Figure 8. The analysis shows (Figure 8a–e) that for the HT0 state, the predominant plane was <101> with the {110} direction, which was the same for the annealed state. Kernel average misorientation documents a significant proportion of small-angle grain boundaries that contour cellular dislocation subgrains. The proportion of small-angle boundaries (<15°) for the condition HT0 was 87%. The high values of the small-angle boundaries also correspond to kernel average misorientation (KAM) maps with a high areal density of small-angle boundaries (KAM<5°), which is documented in Figure 8d.

For a more detailed analysis of the cellular dislocation subgrains, a TEM foil of the sample HT0 was analyzed. The region in the subgrain with the same orientation (Figure 9a) and the region crossing the boundary between two subgrains (Figure 9b) were analyzed on the TEM foils. It is clear from the analyzed areas that within a subgrain with the same orientation, individual deviations in orientation are smaller than KAM<5°. When analyzing two adjacent subgrains, the KAM at the substructure boundary was KAM>5°.

### 3.3. Microstructure Analysis of Samples after Heat Treatment

The microstructure in Figure 10 documents that after the selected solution annealing and subsequent artificial aging, recrystallization of the grains took place with the exclusion of oxide and carbide particles, which are mainly visible in the SEM images. The average grain size after solution annealing and artificial aging was 35 µm. In Figure 10a it is visible as a typical quasi-polyhedral structure of austenite grains of different sizes. At the grain boundaries, it is possible to observe the precipitation of oxide and carbide particles (Figure 10b). These particles contour the grain boundaries as can be seen in Figure 10c. An SEM analysis confirmed a significant increase in precipitates compared to the HT0 condition. Compared to the previous state, HT0, no cellular or columnar dislocation substructures are observed in the SEM images.

The TEM and STEM images clearly show that in the material after the selected heat treatment, the density of the dislocations significantly decreased. On the other hand, the density and size of the precipitates increased. At the same time, there was the formation of large elongated precipitates (tens to hundreds of nm in length), which contoured the high-angle boundaries. Most grain boundaries are contoured by elongated precipitates of Cr_23_C_6_ or in some cases (Mn, Cr_2_)O_4_. The small precipitates distributed inside the grains increased in some cases by up to 100 nm. Two types of precipitates in the grains and two types of precipitates contouring the high-angle grain boundaries were identified using STEM EDS. The large precipitates $\left(>40\;\mathrm{nm}\right)$ inside the grains and along the grain boundaries were of the type (Mn, Cr_2_)O_4_, shown in Figure 11 (spectrum 2). The smaller precipitates $\left(<40\;\mathrm{nm}\right)$ inside the grains were of the MnO_2_ type, shown in Figure 11 (spectrum 4, 5, 6, 8).

Artificial aging at 700 °C and for 50 h relaxed the internal thermal stresses in the material a bit more and additionally promoted the diffusion of elements, which helps in the formation of new precipitates. The subgrain boundaries can still be observed in the structure, but the dislocations between the individual subgrains show a lower density. The dislocations present in the HT2 samples still form small-angle boundaries even after the selected heat treatment. The small-angle boundaries can be seen in Figure 12 and are marked by red arrows.

Large elongated precipitates contouring small-angle grain boundaries are not visible in the HT2 state. In Figure 13, red arrows point to the prominent austenitic nano twins which were observed.

An EBSD analysis was performed on the samples after heat treatment. There is an inverse pole figure in the direction of the *Z*-axis in Figure 14. After the selected heat treatment, a significant decrease in KAM<5° is visible (Figure 14b–d), which is mainly related to the reduced number of dislocations. This results in a reduction of internal stresses in the material. The total number of KAM<15° was approximately the same as in the HT0 condition. The proportion of borders with KAM<15° was 86%. From the analysis, it can be assumed that by reducing the number of dislocations, there is also a more pronounced decrease in KAM<0.5° within the individual cellular subgrains.

## 4. Discussion

### 4.1. Analysis of the Investigated Material

The authors of [53,54] determined that the boundaries of the individual subgrains are formed by a considerable amount of oxide particles containing the elements Mn and Si. This occurrence of oxides based on Mn and Si is also documented in the results presented here. According to these previous works, contamination with oxides can occur especially when using recycled powder from previous production or when the working space of the device is contaminated with oxygen. Works [54,55] demonstrated that the amount of these particles do not have a significant effect on the resulting properties of YS and UTS. According to work [54], Si-based oxide nanoparticles improve mechanical properties. This influence on the mechanical properties, especially strength properties, was also confirmed in this study; no significant deviations were observed during the static tensile tests, and these oxides did not negatively affect the strength properties of the samples. According to paper [56], a metal powder’s quality, purity, and sphericity significantly affect the resulting mechanical properties. Paper [57] documents that oxides can be present in the laser chamber from a previous print. These oxide particles are subsequently drawn into the laser beam due to the high pressure and thus penetrate the liquid phase. According to [56], oxides are already formed at a content of 0.1% O_2_. Several works have dealt with the idea of the in situ formation of oxide-dispersion-strengthened steels using L-PBF technology. Numerous types of research are still ongoing in this area to clarify the formation and effects of oxide nanoparticles in 316L produced by L-PBF technology. Since many studies dealing with deoxidation reactions during solidification in casting and welding processes have been performed, some of this knowledge, such as mechanisms for reducing the formation of oxide particles, can be directly applied to the L-PBF process. Since the manufactured parts must undergo heat treatment, it is important to investigate the evolution of these particles during heat treatment. The precipitates observed in all the samples were equally represented at the boundaries and inside the grains. Larger oxide particles were concentrated preferentially on the grain boundaries. The oxide particles observed in the SEM images and TEM EDS were of SiO_2_ and MnSiO_2_ type. Small amounts of Si are added to 316L steel to bind Mo to increase corrosion resistance. Even at a low Si content, the formation of oxide particles occurs [54]. At a content of $\mathrm{Si}\in \langle 0;\;0.5\rangle \;\left[\%\right]$, the strength properties are slightly affected. At a content of $\mathrm{Si}\in \langle 0.5;\;1.5\rangle \;\left[\%\right]$, there is a sharp decrease in strength and plastic properties. According to work [58], the segregation of oxide particles occurs mainly in the area of the pores. This aspect was not observed in our evaluation of the microstructures. Porosity was <0.1%. According to paper [50], it is possible to claim a negative impact of porosity on mechanical properties only for values of porosity >1%. Considering the content of Si in the analyzed material, which was <0.5%, it can be concluded that the analyzed oxide particles, in combination with a low value of porosity and low content of Si, did not influence the resulting mechanical properties.

### 4.2. Characteristics of the Microstructure after L-PBF

The specific structure arising directly after L-PBF is the main topic of many scientific publications [59,60,61,62,63]. In previous studies [12,15,26,39,53], high-density entangled dislocations were observed at cellular subgrain boundaries, consistent with this study’s observations. As opposed to their conventionally manufactured counterparts, which develop cellular dislocation networks during deformation, the cellular dislocation network of L-PBF-fabricated 316L stainless steel can be stabilized through loading, by both segregated alloying atoms and small misorientations between cells resulting from rapid solidification. Based on several studies [12,15,59], the entangled dislocation networks’ subgrain boundaries will therefore serve as barriers, representing grain boundaries, where dislocations pile up during plastic deformation. In this case, the cellular subgrains have a greater effect on the strengthening of 316L stainless steel than grain boundary strengthening. The cellular substructure of L-PBF specimens can result in significant differences in their mechanical properties, resulting in higher YS. Several studies [31,34,42] have demonstrated that this closely spaced honeycomb structure produces a high dislocation density, which leads to unexpected differences in the mechanical properties of the samples.

Based on the experimental results and the literature review, the formation of the structure and substructure can be described by the model shown in Figure 15. The green line separates individual subgrains with a small misorientation, (KAM) <5°, which was observed in the performed EBSD analyses. Individual subgrains are formed by smaller (200–500 nm) dislocation subgrains. Dislocation subgrains are referred to as cellular dislocation subgrains and can take the form of columnar dislocation subgrains during epitaxial growth. The majority component of the substructure consists of cellular dislocation subgrains. Cellular dislocation subgrains have the same crystallographic orientation within a single subgrain, with KAM<0.5° (Figure 15—green dashed line). Cellular dislocation subgrains are the result of significant thermal events during production (formation of weld-strip boundaries) and the microsegregation of elements at the boundaries of cellular dislocation subgrains. The combination of these parameters affects the growth of cellular dislocation subgrains. According to [62], the kinetics of thermal events during production can be influenced by changes in production parameters. Individual subgrains with the same crystallographic orientation are bounded by high-angle boundaries (red line). High-angle grain boundaries are not affected by weld-band boundaries (blue line) as they are by cellular-dislocation-subgrain boundaries. The resulting structure of the material after L-PBF production is characterized by a high number of dislocations, the density of which can be compared to significantly deformed materials.

### 4.3. Grain Development in L-PBF after Heat Treatment

The purpose of heat treatment was to find a suitable heating temperature and time, e.g., from the point of view of the reduction of internal thermal stresses, as well as the reshaping of the structure, from having the character of a large nucleation surface corresponding to the shape of elongated grains after L-PBF, to having a smaller nucleation surface value approaching the shape of polyhedral grains. Currently, scientific research is focused on explaining the origin and course of dislocation substructure formation, and its further transformation during heat treatment [63,64,65,66].

According to [42,44,47], it is possible to claim that the selected heat treatment and development of grains and subgrains have a significant impact on mechanical properties. In general, this is because after the selected heat treatment there is a significant reduction in the number of dislocations, which results in a decrease in YS and, conversely, an increase in total elongation. This is related to the resulting reduction of internal stresses after the production itself, which are removed by the given heat treatment.

However, HT2 exhibits higher strain hardening resulting in a greater similarity to UTS than HT0.

Based on the performed analyses, it is possible to characterize the growth mechanism of subgrains and grains under different temperature conditions. The transformation of the dislocation substructure changes depending on the thermal influence of the overall structure. This growth is shown schematically in Figure 16 and can be defined as follows:
-Directly after L-PBF, an austenitic structure with a large number of subgrains is formed. The resulting subgrains are made up of a large number of dislocations. Dislocation tangles on the subgrain boundaries are visible on TEM images of the HT0 state (Figure 16). The authors of [25,65,66,67] described the causes of the dislocation substructure based on internal thermal stresses, and stated that L-PBF technology has elements of plastic deformation. The high density of the dislocations and the occurrence of nano twins in the structure confirmed this fact. Plastic deformation as a secondary effect of internal thermal stresses is induced by high-energy laser energy in combination with rapid cooling $\left(\sim 10^{4}\;\mathrm{to}\;10^{6}\;K\cdot s^{-1}\right)$.-After being heated to 1050 °C, dislocations are gradually annihilated. Subgrain dislocation boundaries are still visible at their original locations. The reduction of the dislocation density is visible in the TEM images of the HT2 state. The cellular dislocation subgrains retain their original position, as documented by the TEM images of the HT2 state and the EBSD analysis of the small-angle boundaries compared to the HT0 state. There is an increase in the size of the precipitates from the original of approximately 15 nm to 150 nm and more. After heating to a temperature of 1050 °C, the size of the grains is not affected, and they remain approximately the same size.-Based on works [24,68], it can be concluded that at temperatures of 1200 °C and higher, dislocations are annihilated. In the temperature range of 1200 °C, new subgrains with irregular shapes are formed. The new subgrains are approximately 1 µm in size and continue to be bounded by wide-angle boundaries.-At temperatures higher than 1250 °C, small-angle subgrain boundaries are transformed into large-angle new grain boundaries. Abnormal grain growth may occur during this transformation. At temperatures of 1250 °C and above, the resulting grains have a size of approximately 30–50 µm [28].


## 5. Conclusions

In this paper, the structure formed directly after L-PBF and after the selected heat treatment was analyzed. The summary of the main survey results is as follows:
-After L-PBF, a specific substructure was observed in the structure, which is typical for the given technology. Individual grains were formed by low-angle subgrains, which were formed by cellular dislocation subgrains, with KAM<0.5°. The proportion of low-angle boundaries $\left(\mathrm{KAM}<15{}^{\circ}\right)$ was 87%. The size of the cellular dislocation subgrains was in the interval $\langle 300-700\rangle \;\left[\mathrm{nm}\right]$;-After the selected heat treatment, the proportion of low- and high-angle boundaries did not change. The proportion of low-angle boundaries $\left(\mathrm{KAM}<15{}^{\circ}\right)$ was 86%. The heat treatment did not affect the orientation of the grains, which were preferentially oriented in the ⟨101⟩ plane with the $\left\{110\right\}$ direction;-Heat treatment did not have a significant effect on the size of the individual grains either. For the HT0 state, $D=\langle 25-30\rangle \;\left[µm\right]$, and for the HT2 state, $D=\langle 34;\;35\rangle \;\left[µm\right]$;-TEM EDS analysis pointed to the precipitation of Cr and Ni elements at the boundaries of the cellular dislocation subgrains. The formation of oxide particles MnSiO_2_, CrSiO_2_, (Mn, Cr_2_)O_4_, and MnO_2_, whose dispersion in the structure was random, was monitored. Cr_23_C_6_ carbide particles were concentrated on the grain boundaries;-TEM analyses showed that after the heat treatment HT2, there was a significant reduction in the number of dislocations and an increase in precipitates size by ~100 nm;-From the performed analyses of the mechanical properties of the initial state HT0, and the heat-treated state HT2, at $T_{t}=298\;K,\;77\;K,\;and\;8\;K$, it is possible to observe a significant increase in strength properties when the testing temperature is reduced;-The achieved mechanical and microstructural properties of the 316L stainless steel produced by L-PBF are significant for the field of material science and their use in the energy industry has a high potential.


## Figures and Tables

**Figure 1 materials-16-03935-f001:**
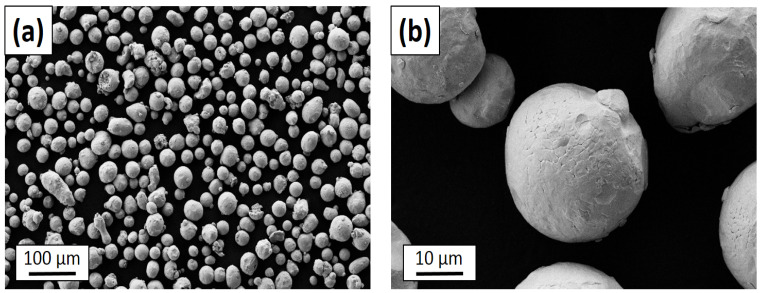
(**a**) The investigated powder material 316L stainless steel and (**b**) detail of a globular particle.

**Figure 2 materials-16-03935-f002:**
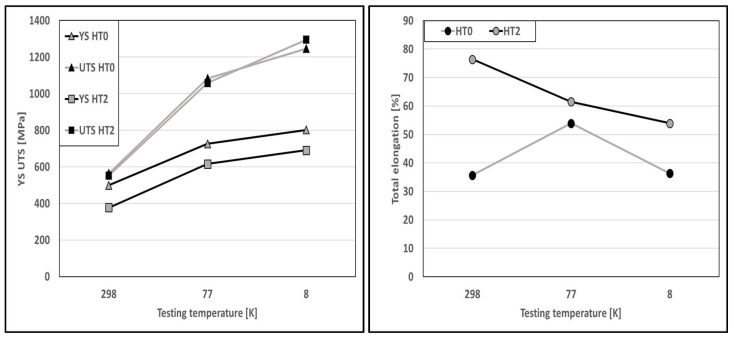
Mechanical properties of 316L stainless steel tested at ambient temperature, 77 K, and 8 K.

**Figure 3 materials-16-03935-f003:**
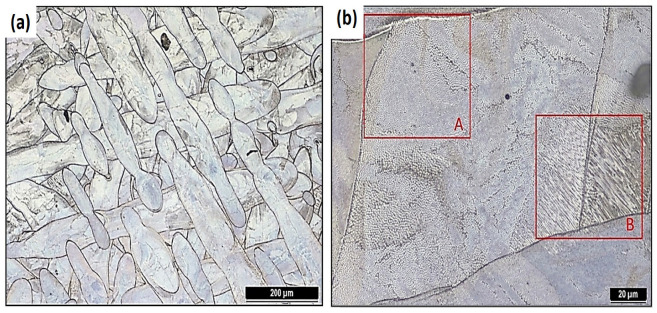
The microstructure of the HT0 state: (**a**) visible weld bands and (**b**) areas with different substructure morphology (A—cellular dislocation type of substructure; B—columnar dislocation type of substructure).

**Figure 4 materials-16-03935-f004:**
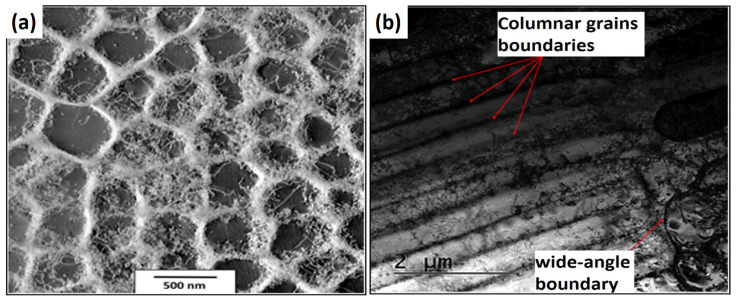
STEM images of HT0 state: (**a**) cellular dislocation subgrains and (**b**) columnar dislocation subgrains.

**Figure 5 materials-16-03935-f005:**
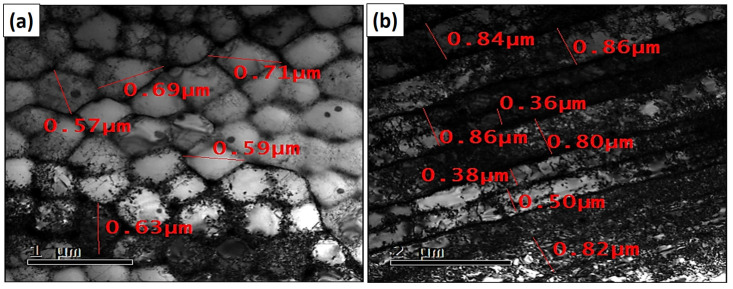
Bright-field TEM images of the measurement dimensions of (**a**) cellular dislocation subgrains and (**b**) columnar dislocation subgrains.

**Figure 6 materials-16-03935-f006:**
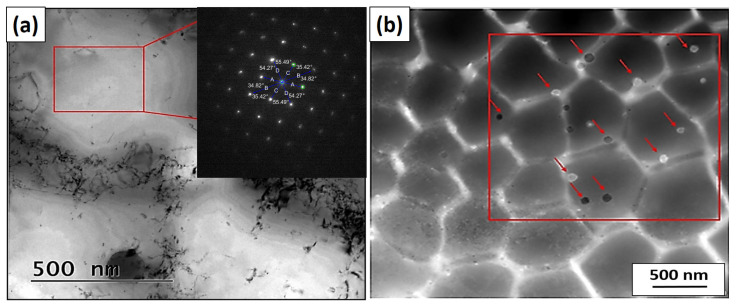
(**a**) TEM using selected area electron diffraction analysis of the γ-Fe matrix (red box) in the axis zone (ZA) [0, −1, 1]. (**b**) Diffraction contrast in bright field STEM images with excluded particles marked in the red box.

**Figure 7 materials-16-03935-f007:**
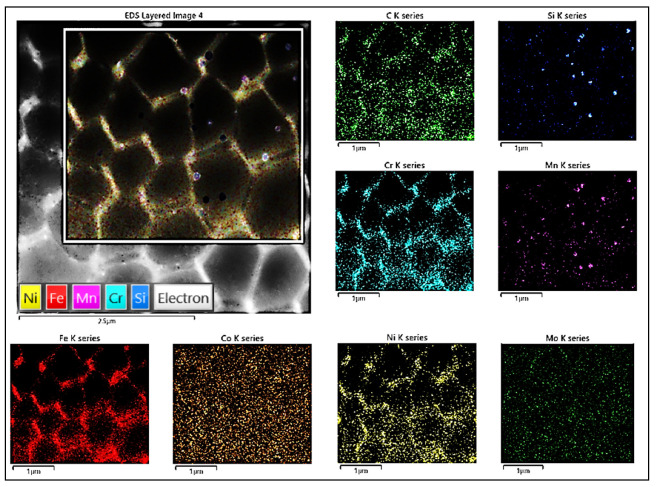
An EDS analysis of the chemical composition of the subgrain boundaries and excluded particles.

**Figure 8 materials-16-03935-f008:**
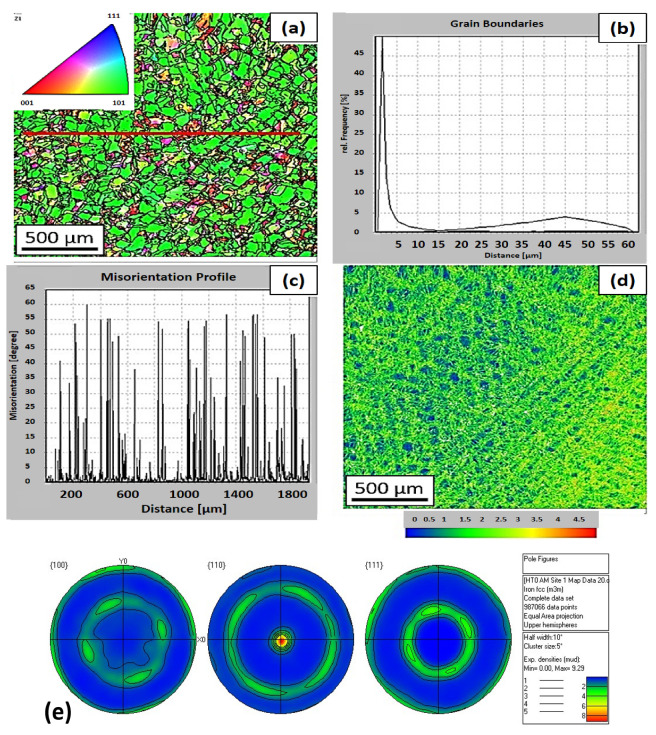
EBSD maps of the HT0 state: (**a**) inverse pole figure (with line analysis) in the transverse direction to the built direction, (**b**) the frequency of Kernel average misorientation, (**c**) the distribution of small- and large-angle boundaries along a straight line, (**d**) kernel average misorientation map, and (**e**) pole figures.

**Figure 9 materials-16-03935-f009:**
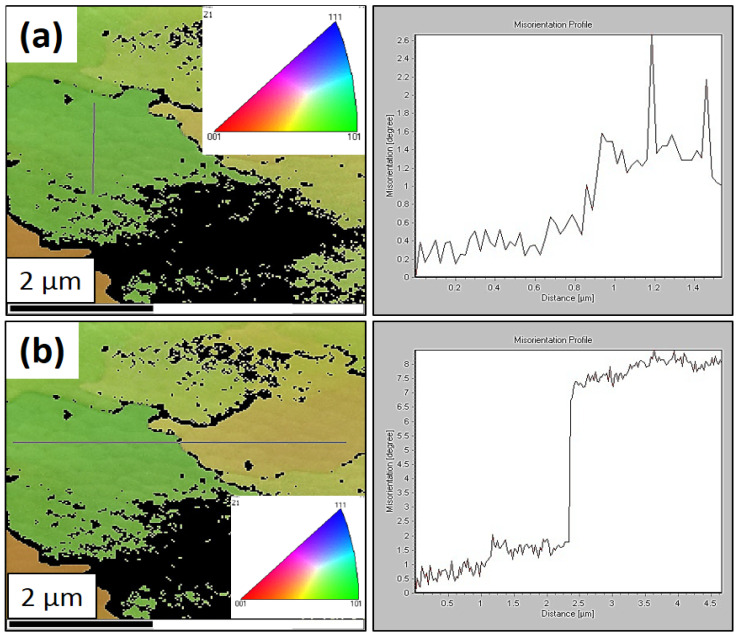
TEM and EBSD maps of the HT0 state: (**a**) inverse pole figure in the transverse direction with a linear analysis (gray line) within one subgrain and the kernel average misorientation profile and (**b**) inverse pole figure in the transverse direction with a linear analysis (gray line) through a subgrain boundary and the kernel average misorientation profile.

**Figure 10 materials-16-03935-f010:**
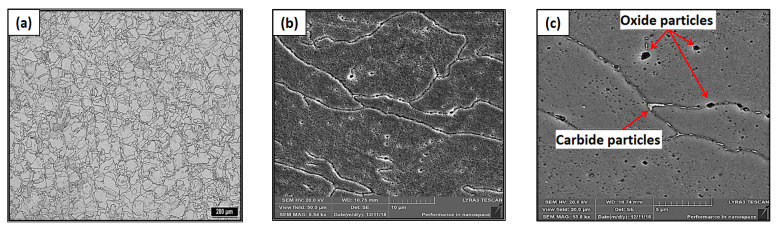
OM and SEM microstructure of HT2 state: (**a**) quasi-polyhedral microstructure of austenite grains, (**b**) visible excluded oxide and carbide particles at the grain boundaries and inside the grain, and (**c**) detailed backscattered electrons (BSE)-SEM analysis—excluded oxidic (black) and carbidic (light) particles contouring the grain boundaries.

**Figure 11 materials-16-03935-f011:**
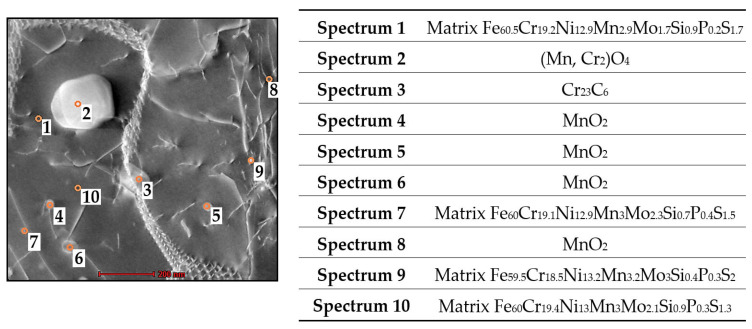
STEM EDS analysis of the nanoparticles and matrix in sample HT2 with spectra analysis.

**Figure 12 materials-16-03935-f012:**
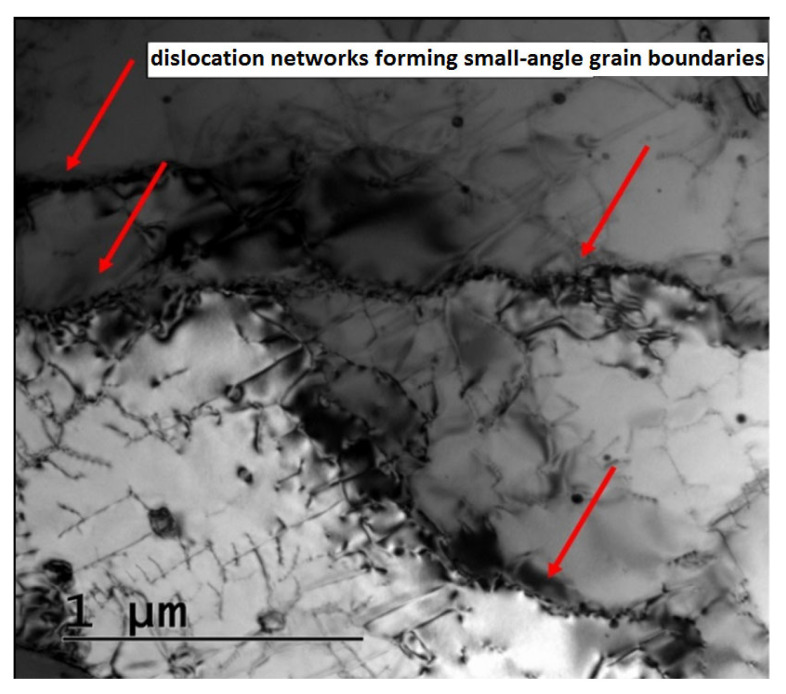
Bright-field TEM images of dislocations forming small-angle boundaries after heat treatment.

**Figure 13 materials-16-03935-f013:**
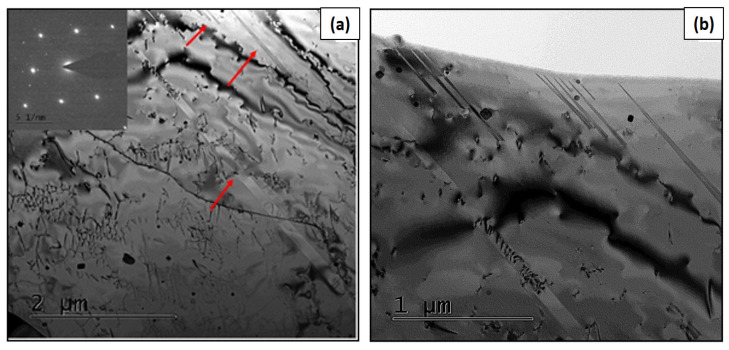
BF TEM images of nanotwins: (**a**) nanotwins inside the sample with diffraction pattern, (**b**) Nanotwins form from the edge of the sample.

**Figure 14 materials-16-03935-f014:**
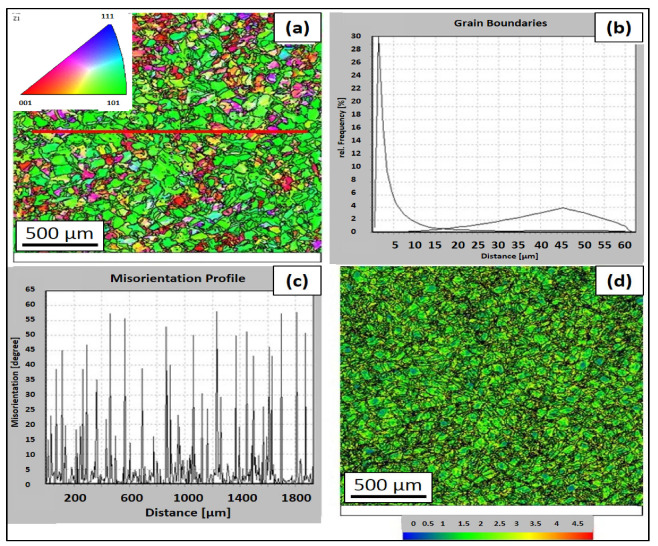
EBSD maps of the HT2 state: (**a**) inverse pole figure in the transverse direction to the built direction, (**b**) the frequency of the kernel average misorientation, (**c**) the distribution of small- and large-angle boundaries along a straight line, and (**d**) the kernel average misorientation map.

**Figure 15 materials-16-03935-f015:**
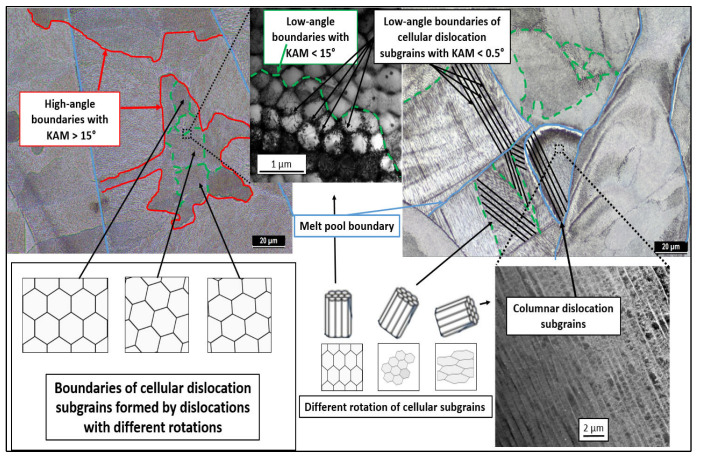
Schematic model of the characteristic structure and substructure created after use of L-PBF technology for austenitic stainless steel 316L for the state without heat treatment (as-built).

**Figure 16 materials-16-03935-f016:**
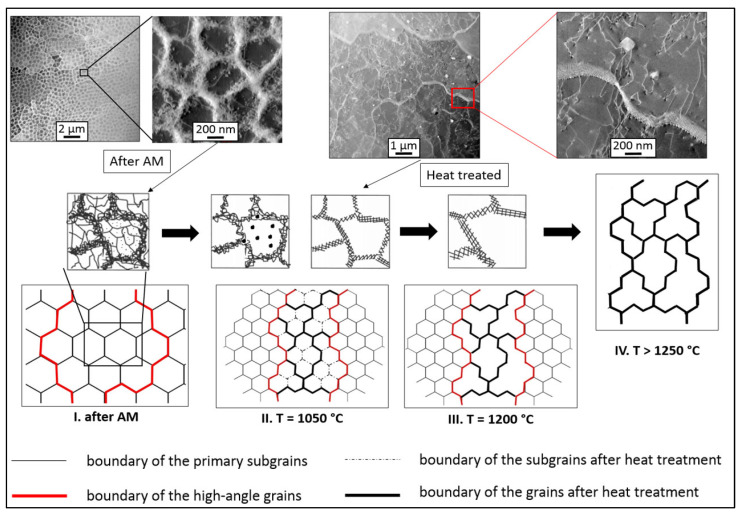
Model of the formation of the structure of the 316L material using the L-PBF technology after the selected heat treatment.

**Table 1 materials-16-03935-t001:** The process parameters for the sample production using L-PBF technology.

*P* (W)	*S_S_* (mm·s^−1^)	*h_d_* (mm)	*S*_1_ (mm)
195	800	0.09	800

## Data Availability

Not applicable.

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
