# Peer review of "Investigation of the Properties of 316L Stainless Steel after AM and Heat Treatment"

_materials, 2023, doi:10.3390/ma16113935_

Round 1

Reviewer 1 Report

well done

Author Response

The authors thank reviewer 1 for the time and effort that the reviewer has dedicated to provide valuable feedback on our manuscript.

Reviewer 2 Report

 The paper investigates the microstructure and mechanical properties of 316L stainless steel prepared by additive manufacturing and heat treatment. The study is well-conducted and provides valuable insights into the effect of heat treatment on the microstructure and properties of the material. However, there are some points that could be improved:

 1- The abstract should provide a more concise and focused summary of the study, highlighting the main findings and their significance.

2- The introduction should provide more context and background information on the importance and relevance of studying the microstructure and properties of 316L stainless steel prepared by additive manufacturing.

3- The paper should provide a comprehensive review of the relevant literature, and how the current study contributes to this field.

4- The paper should clearly state the research questions or hypotheses that guided the study, and how the study addressed them. This will help readers understand the purpose of the study and the significance of the findings.

5- The materials and methods section should provide more detail on the experimental procedures and techniques used in the study, including the parameters of the laser powder bed fusion process, the heat treatment conditions, and the testing methods for mechanical properties.

6- The discussion section should go beyond a simple summary of the results and provide a critical analysis and interpretation of the findings, including their implications for future research and practical applications.

7- The paper should provide more detail on the limitations and potential sources of error in the study, and how these were addressed or minimized.

8- The authors should consider expanding the scope of the study to include more materials and testing conditions, in order to provide a more comprehensive understanding of the microstructure and properties of 316L stainless steel prepared by additive manufacturing.

9-While the authors briefly mention the potential advantages of AM for producing materials with comparable properties to conventional technologies, a more detailed discussion of the current state of research in this area would provide a stronger foundation for the study. Additionally, while the conclusions drawn from the data are reasonable, the authors could provide more discussion of the implications of their findings and how they contribute to the broader field of materials science.

10- The paper should be proofread and edited carefully to ensure clarity, accuracy, and consistency in language and formatting.

The paper should be proofread and edited carefully to ensure clarity, accuracy, and consistency in language and formatting.

Author Response

We are thankful to reviewer 2 for constructive comments and suggestions on our paper. We appreciate the time and effort that reviewer 2 has dedicated to providing valuable feedback on our manuscript. We have incorporated the changes to reflect most of the suggestions provided by reviewer 2.

Modification made in revised manuscript are marked up.

Comment 1: The abstract should provide a more concise and focused summary of the study, highlighting the main findings and their significance.

Answer: In the abstract, information was added about the results: grain size, subgrain size, and precipitate size, which were obtained within the scope of this study.

Comment 2: The introduction should provide more context and background information on the importance and relevance of studying the microstructure and properties of 316L stainless steel prepared by additive manufacturing.

Answer: Information on using 316L stainless steel for ITER was added in the introduction. The mechanical properties were specified in more detail depending on the use. The effect of heat treatment was also described. Please see lines: 56-58; 85-87; 90-95. The introduction was supplemented with information that informs about the methodologies used. Please see lines: 98-102.

Comment 3: The paper should provide a comprehensive review of the relevant literature, and how the current study contributes to this field.

Answer: The manuscript is conceived based on a current literature survey. The total number of literary sources is 66, of which a large part is discussed in the discussion chapter.

Comment 4: The paper should clearly state the research questions or hypotheses that guided the study, and how the study addressed them. This will help readers understand the purpose of the study and the significance of the findings.

Answer: The manuscript deals with the properties of 316L stainless steel produced by AM technology, namely L-PBF. The main idea of the manuscript is to follow the development of material structures after production and selected heat treatment. The specificity of the materials produced in this way is their unique structure, which is formed predominantly by cellular dislocation subgrains. This evolution of dislocation subgrains (under selected heat treatment) is the main output of the two models presented in Figures 16 and 17. The development and hypothesis are presented in the introduction in lines 72-78; 85-89, and 90-92.

Comment 5: The materials and methods section should provide more detail on the experimental procedures and techniques used in the study, including the parameters of the laser powder bed fusion process, the heat treatment conditions, and the testing methods for mechanical properties.

Answer: The following information was added in the materials and methods section:

  • the image of investigated powder material was added (Figure 1)
  • information about TEM microscopes on which structural analyzes were carried out with a description of samples preparation was added. Please see lines 146-156.
  • The procedure for preparing TEM samples was added. Please see lines 146-147.
  • For the preparation of samples by L-PBF, a standard procedure resulting from the options defined for the device was used.
  • conditions of heat treatment (HT2) are given in the chapter: "Heat treatment and microstructure"
  • the conditions of the static tensile test are given in the chapter: "The static tensile test". The samples were tested on two devices. On the MTS 100 Landmark machine, the samples were tested at cryogenic temperatures, and for tests at ambient temperatures, they were tested on the TINIUS OLSEN machine. The tests were carried out in accordance with ASTM E8M.

Comment 6: The discussion section should go beyond a simple summary of the results and provide a critical analysis and interpretation of the findings, including their implications for future research and practical applications.

Answer: The discussion offers a comprehensive view of the development of the structure from the powder material used, through the production process to the selected heat treatment. The discussion was supplemented with additional information that better characterizes the resulting structure. The discussion was supplemented with an evaluation of the mechanical properties about the structure after L-PBF

Comment 7: The paper should provide more detail on the limitations and potential sources of error in the study, and how these were addressed or minimized.

Answer: The manuscript has the character of basic research, which does not result in a technological application. From the point of view of knowledge, the resulting models have a significant contribution, which points to the development of the substructure, which has an impact on the mechanical properties.

Comment 8: The authors should consider expanding the scope of the study to include more materials and testing conditions, in order to provide a more comprehensive understanding of the microstructure and properties of 316L stainless steel prepared by additive manufacturing.

Answer: The submitted manuscript deals exclusively with the material 316L stainless steel. The author's team considers the methodology used to be sufficient for evaluating the analyzed properties. Methods used: particle analysis, L-PBF technology, optical microscopy, scanning electron microscopy (EBSD), transmission electron microscopy (EDS, TEM EBSD). To obtain mechanical properties, static tensile tests were performed at 8K, 77K, and ambient temperature.

Because the editing of manuscript is limited, we are preparing a new article which will discussed given questions.

Comment 9: While the authors briefly mention the potential advantages of AM for producing materials with comparable properties to conventional technologies, a more detailed discussion of the current state of research in this area would provide a stronger foundation for the study. Additionally, while the conclusions drawn from the data are reasonable, the authors could provide more discussion of the implications of their findings and how they contribute to the broader field of materials science.

Answer: The discussion part of the manuscript was supplemented. Please see lines 378-391, and 408-411, and 424-430. Into the manuscript was added the EBSD analysis that complements and explains the information from the discussion. In conclusion, the contribution to the field of material science was defined.

Comment 10: The paper should be proofread and edited carefully to ensure clarity, accuracy, and consistency in language and formatting.

Answer: The manuscript has been corrected. Slightly visible scales have been fixed. Language errors have been corrected. Typos and other errors have been removed.

Reviewer 3 Report

The authors conducted an experimental investigation on the mechanical performance and microstructural evolution of 316L stainless steel by laser powder bed fusion (L-PBF) technique and selected heat treatment. The samples directly prepared through the additive manufacturing method (L-PBF) demonstrate similar mechanical performances including ultimate tensile strength. Low-angle and high-angle subgrains are identified through SEM, and TEM characterizations. With the selected heat process, the number of dislocations is reduced, and the size of precipitates is improved. Such behavior has been qualitatively explained by two grain evolution models for the L-PBF process and post-heat treatment.  

The manuscript is well-organized overall, and the topic is related to the interest of this journal. However, before this manuscript becomes acceptable, the authors are required to address the following comments well.

1, What does “PM” represent in the title of this manuscript?

2, In the abstract (line 24), the statement about the precipitates should be precise. It’s an increase in precipitate size.

3, In line 94, the author is trying to use Figure 1 to show the morphology of the powders. However, Figure 1 provided in the manuscript is related to mechanical performance. Please add the figure illustrating the powders.

4, In line 108, is the atmosphere filled with Ar or Ag? If it’s Ar, please modify the statement.

5, What is the thickness of TEM samples? And what’s the procedure for preparing TEM samples?

6, In line 145, there are two “however” in one sentence.

7, In Figure 1, what does A5 represent?

8, Please explain why the samples have reduced yield strength after heat treatment?

9, In line 163, “Figure 2a-area A” should be “Figure 2b-area A”.

10, There are lots of acronyms without explaining the full name in the manuscript. Please find the following items and add the full name at the first appearance.

      OM, STEM, BF, BSE, LM, EBSD, EDS, SAED

11, In line 179, it should be “STEM” rather than “SEM”.

12, In Figure 3a, what are the fiber-like structures on the boundary of cellular subgrains?

13, In Figure 4, there is an additional SPACE in the caption between “of” and “dimensions”.

14, In line 206, please provide the analysis of how the diffraction contrast of cellular dislocation indicates the segregation of Ni and Cr. Based on the EDS mapping, it cannot be confirmed that Ni and Cr are segregated.

15, In Figure 7, there is no figure labeled as a, b, or c.

16, In line 305, please add figures showing EBSD results and analysis.

17, In Figure 11, “0, 5” should be “0.5”.

18, Please remove the red wavy lines under the statements in Figure 11 and Figure 12.

19, The scale bar of Figure 5b and Figure 12 are not clear.

20, Much information is only provided in the conclusion part, such as the orientation of grains. Please add detailed discussions in the manuscript.

21, In line 387, what does Ttest represent?

Based on the abovementioned comments, this manuscript is recommended for major revision. A revised version is required.

Author Response

We are thankful to reviewer 3 for constructive comments and suggestions on our paper. We appreciate the time and effort that reviewer 3 has dedicated to providing valuable feedback on our manuscript. We have incorporated the changes to reflect most of the suggestions provided by reviewer 3.

Modifications made in the revised manuscript are marked.

Comment 1: What does “PM” represent in the title of this manuscript?

Answer: PM in the title of the manuscript represents "powder metallurgy". After consideration, the abbreviation PM was removed from the title of the manuscript.

Comment 2: In the abstract (line 24), the statement about the precipitates should be precise. It’s an increase in precipitate size.

Answer: The statement was edited. Please see lines 24-30.

Comment 3: In line 94, the author is trying to use Figure 1 to show the morphology of the powders. However, Figure 1 provided in the manuscript is related to mechanical performance. Please add the figure illustrating the powders.

Answer: Figure 1 was added to the manuscript. Please see line 121.

Comment 4: In line 108, is the atmosphere filled with Ar or Ag? If it’s Ar, please modify the statement.

Answer: Ar atmosphere is correct. The error was removed and the statement was corrected. Please see line 141.

Comment 5: What is the thickness of TEM samples? And what’s the procedure for preparing TEM samples?

Answer: A more accurate sample preparation procedure for TEM was added. Please see lines 146-155.

Comment 6: In line 145, there are two “however” in one sentence.

Answer: The sentence was edited and one "however" was removed. Please see line 173.

Comment 7: In Figure 1, what does A5 represent?

Answer: A5 is the ratio of the original specimen gauge length to diameter ratio of 5. In Figure 2, A5 was changed to total elongation.

Comment 8: Please explain why the samples have reduced yield strength after heat treatment?

Answer: The explanation of why the YS reduction occurs after heat treatment was added to the discussion. Please see lines 424-430.

Comment 9: In line 163, “Figure 2a-area A” should be “Figure 2b-area A”.

Answer: The incorrect marking in the text has been corrected. Please see line 190.

Comment 10: There are lots of acronyms without explaining the full name in the manuscript. Please find the following items and add the full name at the first appearance.

      OM, STEM, BF, BSE, LM, EBSD, EDS, SAED

Answer: Explanations of acronyms were incorporated into the manuscript.

Comment 11: In line 179, it should be “STEM” rather than “SEM”.

Answer: The abbreviation SEM was replaced by STEM. Please see line 206.

Comment 12: In Figure 3a, what are the fiber-like structures on the boundary of cellular subgrains?

Answer: The boundaries of individual cellular subgrains are formed by significant clusters of dislocations, which are typical for substructures created after the L-PBF technology. The statement was added. Please see lines 210-211.

Comment 13: In Figure 4, there is an additional SPACE in the caption between “of” and “dimensions”.

Answer: Double-space was deleted.

Comment 14: In line 206, please provide the analysis of how the diffraction contrast of cellular dislocation indicates the segregation of Ni and Cr. Based on the EDS mapping, it cannot be confirmed that Ni and Cr are segregated.

Answer: The statement has been removed and replaced with another statement. Please see lines 243-245.

Comment 15: In Figure 7, there is no figure labeled as a, b, or c.

Answer: Marking of images a, b, c was added. Please see Figure 10.

Comment 16: In line 305, please add figures showing EBSD results and analysis.

Answer: The EBSD analysis was added to the manuscript with analysis for the as-built state and after heat treatment. A more detailed TEM EBSD analysis of the as-built state was also added to characterize the structure inside the dislocation subgrains.

Comment 17: In Figure 11, “0, 5” should be “0.5”.

Answer: In Figure 15 a comma was replaced by a dot.

Comment 18: Please remove the red wavy lines under the statements in Figure 11 and Figure 12.

Answer: The red wavy lines were deleted in Figure 15 and Figure 16.

Comment 19: The scale bar of Figure 5b and Figure 12 are not clear.

Answer: The scale bars were edited and now are clear.

20, Much information is only provided in the conclusion part, such as the orientation of grains. Please add detailed discussions in the manuscript.

Answer: Some of the results were incorporated into the manuscript in more detail, especially the EBSD analysis, which explains the information from the conclusion, especially grain orientation.

21, In line 387, what does Ttest represent?

Answer: The abbreviation was deleted and it was replaced by „testing temperature“. Please see line 489.

Round 2

Reviewer 2 Report

Accept in present form

Moderate editing of English language

Author Response

I want to thank reviewer 2 for the valuable comments that will improve the quality of the given paper.

Reviewer 3 Report

The authors have addressed the comments well. Just correct the following typos:

1, In line 155, “HClO2” should be “HClO2”.

2, Please remove the second dot in line 255.

3, The following sentence is confusing:

“The smaller size of cellular sub-grains makes them more effective at strengthening 316L stainless steel fabricated by L-PBF than grains”. Please clarify what type of grains you are using as the comparison.

Based on the abovementioned comments, this manuscript is recommended for minor revision. A revised manuscript is required. 

Author Response

We are thankful to reviewer 3 for constructive comments and suggestions on our paper. We have incorporated the changes to the suggestions provided by reviewer 3.

Modifications made in the revised manuscript are marked.

Comment 1: In line 155, “HClO2” should be “HClO2”.

Answer: The index in HClO2 has been corrected.

Comment 2: Please remove the second dot in line 255.

Answer: The second dot was removed from the text.

Comment 3: The following sentence is confusing:

“The smaller size of cellular sub-grains makes them more effective at strengthening 316L stainless steel fabricated by L-PBF than grains”. Please clarify what type of grains you are using as the comparison.

Answer: The statement was edited. Please see lines 388-389.

The statement is linked to the previous statement (lines 384-386). Based on scientific studies and obtained mechanical properties of the given material, it is possible to claim that one of the significant increases in strengthening is also dislocation strengthening, which is the result of the production of the as-built state. This type of strengthening is even more influential than grain boundary strengthening. After the selected heat treatment, the density of dislocations is significantly reduced, and this effect disappears.

I would like to thank reviewer 3 for the valuable comments that will improve the quality of the given paper.
